



# Forced motion simulations of vortex-induced vibrations of wind turbine blades - A study of sensitivities

Christian Grinderslev[1], Felix Houtin-Mongrolle[2], Niels Nørmark Sørensen[3], Georg Raimund Pirrung[3], Pim Jacobs[2], Aqeel Ahmed[4], and Bastien Duboc[4]

[1]Department of Wind and Energy Systems, Technical University of Denmark, 2800, Kgs. Lyngby, Denmark
[2]Siemens Gamesa Renewable Energy, Prinses Beatrixlaan 800, 2595BN Den Haag, Netherlands
[3]Department of Wind and Energy Systems, Technical University of Denmark, Risø Campus, 4000, Roskilde, Denmark
[4]Siemens Gamesa Renewable Energy, 685 Avenue de l'Université, Saint-Etienne-du-Rouvray, 76801, France

**Correspondence:** Christian Grinderslev (cgrinde@dtu.dk)

**Abstract.** Vortex-induced vibrations on wind turbine blades are a complex phenomenon not predictable by standard engineering models. For this reason, higher fidelity computational fluid dynamics (CFD) methods are needed. However, the term CFD covers a broad range of fidelities, and this study investigates which choices have to be made when wanting to capture the VIV phenomenon in a satisfying degree. The method studied is the so-called forced motion (FM) approach, where the structural motion is imposed on the CFD blade surface through modeshape assumptions rather than fully coupled two-way fluid structure interaction. In the study, two independent CFD solvers, EllipSys3D and Ansys CFX, are used and five different turbulence models of varying fidelities are tested. Varying flow scenarios are studied with respectively low to high inclination angles, which determine the component of the flow in the spanwise direction. In all scenarios, the cross-sectional component of the flow is close to perpendicular to the chord of the blade. It is found that the low and high inclination angle scenarios, despite having a difference equivalent to up to only thirty degrees azimuth, have quite different requirements of both grid resolution and turbulence models. For high inclination angles, where the flow has a large spanwise component from tip towards root, satisfying results are found from quite affordable grid sizes, and even with URANS k-ω turbulence the result is quite consistent with models resolving more of the turbulent scales. For low inclination, which has a high degree of natural vortex shedding, the picture is opposite. Here, even for scale resolving turbulence models, a much finer grid resolution is needed. This allows to capture the many incoherent vortices shed from the blade, which have a large impact on the coherent vortices, which inject power into the blade or extract power.

It is found that a good consistency is seen using different variations of the higher fidelity hybrid RANS/LES turbulence models, like IDDES, SBES and k-ω SAS models, which agree well for various flow conditions and imposed amplitudes.

This study shows that extensive care and consideration are needed when modelling 3D VIVs using CFD, as the flow phenomena, and thereby solver requirements, rapidly change for different scenarios.

## Acronyms

- CFD - Computational Fluid Dynamics





- – VIV - vortex-induced Vibration

- – FM - Forced Motion

– FSI - Fluid-Structure Interaction

- – LES - Large Eddy Simulation

- – IDDES - Improved Delayed Detached Eddy Simulation

- – URANS - Unsteady Reynolds-Averaged Navier-Stokes

- – SAS - Scale-Adaptive Simulation

– SBES - Stress-blended eddy simulation

- – QUICK - Quadratic Upstream Interpolation for Convective Kinematics

- – AoA - Angle of attack

- – DTU - Technical University of Denmark

- – SGRE - Siemens Gamesa Renewable Energy

– IEA - International Energy Agency

- – PGL - Parametric Geometry Library

- – VCO - Vertex-Centered-Orthogonality

vortex-induced vibrations (VIVs) on wind turbine blades are a phenomenon gaining relevance as wind turbines become larger and more flexible. When the turbine is not in operation, due to for instance grid loss, maintenance, in storm conditions or

40 during erection, the blades can see wind from various directions, which can result in large angles of attack close to perpendicular to the chord. In this range of wind directions, deep stall with a high degree of vortex shedding can occur, meaning that a risk of lock-in between structural modes and shedding frequencies increases.

As VIVs are directly depending on vortex shedding frequency and phase between the corresponding loads and motion velocity, engineering models struggle to compute the phenomenon. It becomes especially problematic as the blade shape, by

45 twist and chord length, changes over the blade span, making a simple Strouhal relationship analysis difficult. For this reason, high fidelity methods such as computational fluid dynamics (CFD) are needed. Examples of this are the works of Horcas et al. (Horcas et al., 2022a, 2020, 2022b), who studied wind turbine blade VIVs through fluid-structure interaction (FSI) simulations coupling CFD with a structural solver. It was shown that branches of VIVs can be found for various flow angles, defined by so-called pitch and inclination angles. This flow direction definition is also used in the present study and is depicted in Figure

1. As seen, pitch angle, $P$, is the angle between flow and the chord of the root airfoil section. The pitch angle is defined similar



to a standard geometrical angle of attack, reaching $P=90°$ when the wind vector strikes the pressure side perpendicularly to the chord at the root. Inclination, $I$, is the relative vertical angle between the inflow wind and the plane intersecting the root section (i.e. the x-z plane in Fig. 1). $I$ is positive when the spanwise component of the wind flow is from tip to root, and zero when the wind strikes the blade perpendicularly to the span. It is important to notice that various combinations of wind turbine

settings, i.e. blade pitch, yaw and azimuth, can result in identical inclination and pitch angles, meaning these parameters are more general than a single turbine setting.

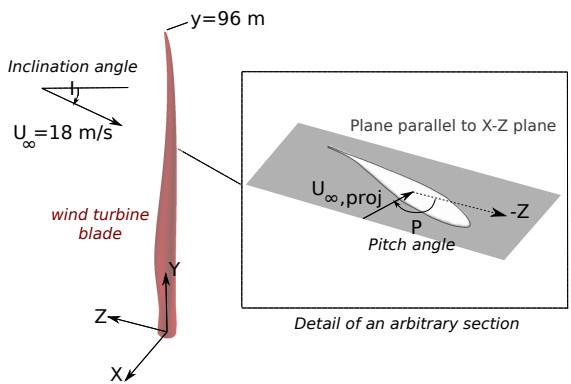

**Figure 1.** Definition of inclination and pitch angles. Reproduced from Horcas et al. (2022b).

The positions of the VIV branches depend on blade shape, structural properties and flow velocity. As shown in (Horcas et al., 2022a, 2020), changing the shape of the blade by tip and/or flap altercations moves the regions of VIVs, however, it does not seem plausible, within realistic alternations, to remove the VIV risk entirely, as the branches rather seem to shift towards other

flow angles and/or velocities.

In a recent study, Grinderslev et al. (2022) used the same setup as Horcas et al. and showed that it is feasible, to omit the coupling with the structural solver, and replace it by an analytical imposed motion of the structural mode. At least when considering single wind turbine blades which are rigidly clamped at the root. The approach is to simulate the forced motion for various defined amplitudes revealing a picture of the power injection by the aerodynamic loads. By comparison to the dissipated

power from structural damping an equilibrium vibration state can then be estimated. The benefit of such an approach is that no coupling framework is needed, and for a specific simulation the approach is likewise faster, as no time for build-up of vibration is needed. This latter benefit, however, disappears when multiple amplitude simulations are needed if the vibration development is of interest.

The forced motion method is no new concept, but has been used extensively on especially cylinder VIVs (Placzek et al.,

2009; Viré et al., 2020) but also for airfoils in 2D (Skrzypiński et al., 2014; Hu et al., 2021).

In the present study, the approach of forced motion (FM) CFD analysis of VIVs is studied further in various aspects. The influence of modelling schemes, turbulence models, grids and more is studied using two well established CFD solvers; one used and developed by the Technical University of Denmark (DTU), EllipSys3D (Sørensen, 1995; Michelsen, 1992), and one





commercial code used by Siemens Gamesa Renewable Energy (SGRE) Ansys CFX (v. 2021, R1, CFX (2021)). The aim is to provide knowledge about good practices when simulating VIVs for wind turbine blades. The present work uses the IEA10MW blade (Bortolotti et al., 2019) also studied in the aforementioned FSI studies by Horcas and Grinderslev et. al.

The study shows that the chosen modelling approach has large effects on the computed power input for cases with low to medium inclination angles, where uncorrelated natural shedding occurs. For cases with high inclination angles, the sensitivity is found to be much lower, as the defined blade motion controls the flow pattern more in this region.

# 1 Methodology

As two different codes are used, variations of grid methods, convective schemes, turbulence models and much more can be studied. Common in all simulations is the use of forced motion CFD simulations as described below in subsection 1.1. In the following subsections, the fluid solver codes will be described along with the chosen models. Finally, the analysis methods will be described.

## 1.1 Forced motion method

In order of doing high-fidelity modelling of VIVs without using a structural coupling framework, the forced motion method (FM) is used. Here, it is assumed that the structural response of the blade seen during VIVs can be simplified to being purely the structural modes. In these simulations, the first edgewise mode has been chosen, as this is the mode being provoked by the investigated flow scenarios when using fully coupled FSI simulations (Horcas et al., 2022b; Grinderslev et al., 2022). In these studies the assumption of having close to purely structural modes has been validated for wind speeds of 18m/s. For high wind speeds, the assumption might not hold as the aeroelastic mode shape moves away from the structural one.

The aeroelastic model of the IEA10MW wind turbine is publicly available from Bortolotti et al. (2019) and in the present study the aeroelastic stability tool HAWCSTAB2 (Hansen, 2004) was used on said model to extract the modes. The first edgewise mode has a frequency of 0.67Hz and the edge and flapwise motion components of the modeshape are depicted in Figure 2, along with the corresponding polynomials fitted for use with the CFD solvers. The phase between flapwise and edgewise amplitude is such that the maximum tip deformation towards the pressure side of the blade occurs at the same time as the maximum deformation towards the leading edge. The amplitude used in the present study is 1m in edgewise direction, except for amplitude sweeps presented in Sec.2.3.

For the specific study, some assumptions are made to enable the FM approach. Firstly, as mentioned, it is assumed that the structural first edgewise mode shape is the only motion present. This means that no contribution from static loads nor buffeting loads are included in the motion. This assumption aligns well with what was found in previous studies using fully coupled FSI (Horcas et al., 2022b; Riva et al., 2022). Another assumption made, is the disregard of the torsional part of the mode shape. This assumption is done for practical reasons in terms of imposing motion in the CFD solvers. The effect has again been tested with FSI simulations which include torsion, and it was found that the consequence of not considering torsion is marginal. In the present case, the torsional component is less than half a degree at the tip for a 1m amplitude. This is, however, not a





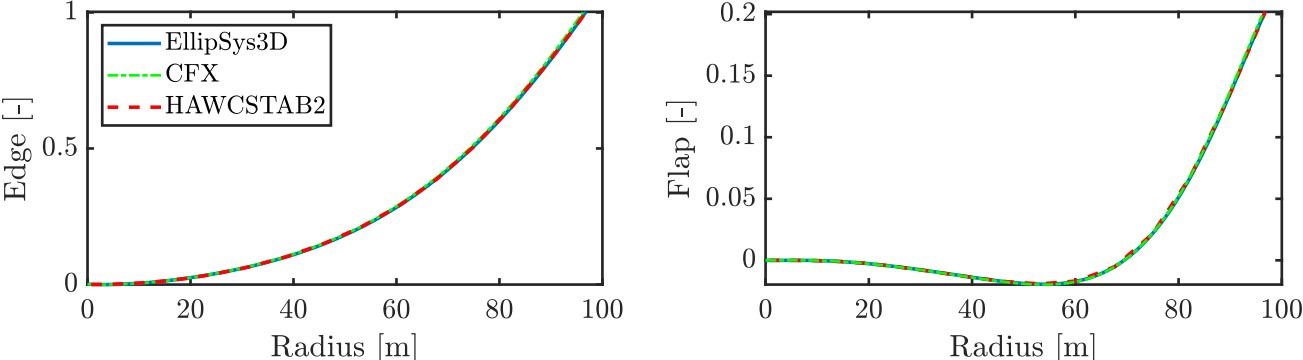

**Figure 2.** First edgewise mode shape of blade from HAWCSTAB2 along with polynomial fit used in forced motion simulations. The effect of the torsional component, which was less than 0.5 degree at the tip for 1 m edgewise deflection, was found to be negligible in Grinderslev et al. (2022). The maximum edgewise deflection to the leading edge occurs together with the maximum flapwise tip deflection towards the pressure side.

.

general conclusion, and is something that should be assessed for the specific blade and flow scenario considered. One reason that torsion has little effect in the present study is the angles of attack (AoA) studied, which are close to perpendicular to the chord. Here, the aerodynamics are less sensitive to small changes of AoA than for instance at stall onset.

## 1.2 EllipSys3D setup

The EllipSys3D CFD solver (Sørensen, 1995; Michelsen, 1994, 1992), is a finite volume code based on structured grids, that solves the incompressible Navier Stokes equations with RANS, LES or hybrid turbulence equations. The solution algorithm is based on the SIMPLE algorithm in combination with Rhie-Chow interpolation to avoid pressure decoupling.

In this study, simulations are based on unsteady Reynolds averaged Navier Stokes (URANS) $k - \omega$ SST (Menter, 1993) along with the higher fidelity $k - \omega$ based improved delayed detached eddy simulations (IDDES) (Gritskevich et al., 2012) for 115 better resolution of turbulent structures shed from the blade.

For the URANS simulations the QUICK convective scheme is used, while for IDDES a combination of QUICK (in RANS region) and fourth order central difference (in LES region) is used as described by Strelets (2001).

### 1.2.1 EllipSys3D Grids

Various grids have been tested in the present study. All surface grids are based on the DTU in-house Parametric Geometry 120 Library (PGL) tool (Zahle, 2022), and volume grids are created through hyperbolic extrusion from the surface grid to a spherical domain with a radius of $\approx 700$m ($\approx 7$ blade lengths), using the mesh tool HypGrid3D (Sørensen, 1998). Multiple grid refinements have been tested to study the sensitivity of the VIVs to the resolved vortices. The baseline mesh used if otherwise





not stated, has 512 cells spanwise along the blade, 256 cells chordwise and 256 cells normal to the surface.The total number of
cells for the baseline mesh is 35.6M cells. This mesh is finer than that used in previous publications (Horcas et al., 2020, 2022b;
Grinderslev et al., 2022), which in this study was found necessary for certain flow scenarios, see Section 2.1. The first cell size
normal to the surface is set to 1e-6m which ensures a y+ of less than one.

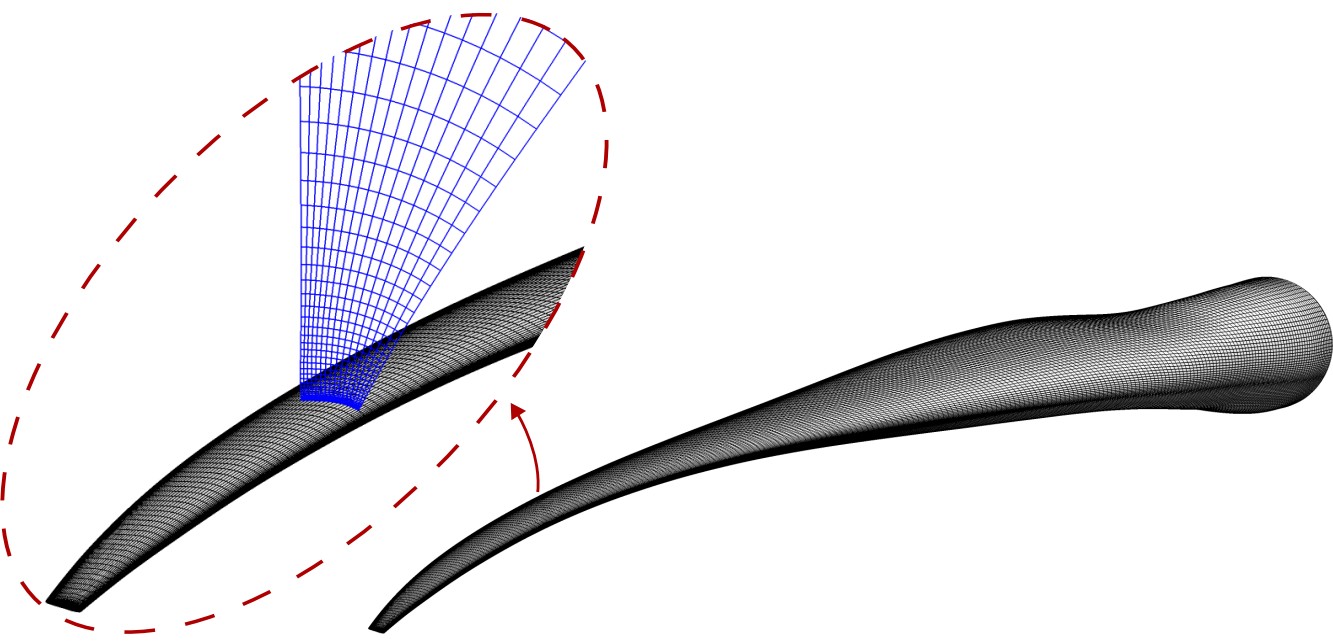

**Figure 3.** Surface grid and volume mesh hyperbolically grown from surface for EllipSys3D. Only every second line is shown for clarity

The grid deformation procedure in the EllipSys3D simulations is based on an explicit algebraic algorithm, transferring the
deformation of the surface grid into the volume grid by a blending approach exploring the block structured nature of the
computational grid.

The deformed grid is computed by enforcing the Cartesian translation and deformation of the surface grid points along the
grid lines normal to the surface. To avoid generating highly non-orthogonal grids at the surface, the normal grid lines are rotated
according to present surface normal direction. Using blending functions in the direction normal to the surface it is assured that
the grid translation and rotation are only enforced in the proximity of the surface of the geometry. This ensures that the original
grid quality is conserved at the surface, while preserving the original grid far away from the surface. In between the surface
proximity region and the far-field region, a blending region is present where the grid quality risks deterioration in the case of
large deformations if the blending is not adequately tuned. Typically, the surface deflections are enforced far away from the
surface, while the rotations are limited to a region close to the surface. The blending is based on hyperbolic tangent functions,
using the normalized curve-length along the grid lines normal to the surface. The procedure can easily be tuned for specific
cases by calibrating the blending function constants for a severe static deformation using a steady state computation.



To assure that the grid is not degenerating, a simple check for negative cells volumes is performed after each grid deformation. No checks for orthogonality are performed.

## 1.3    Ansys CFX setup

The Ansys CFX library (CFX, 2021) gathers a set of solvers to resolve different multi-physical fluid dynamics. Only the incompressible Navier Stokes equations with URANS and hybrid URANS/LES formulation are used in this work. Ansys CFX
is a finite volume code based on structured, unstructured, and hybrid grids. Ansys CFX uses a coupled solver (CFX-Solver, 2021) combined with Rhie-Chow interpolation, which solves the hydrodynamic equations as a single system differing from the SIMPLE algorithm. The solver uses a Multigrid accelerated Incomplete Lower Upper (Raw, 1996) factorization technique for solving the discrete system of linearized equations. It relies on an iterative process to approach the exact solution.

In this study, the simulations are based on URANS with a k-$\omega$ SST (Menter, 1993), SST Scale Adaptive Simulation
(SAS)(Egorov et al., 2010), and Stress Blended Eddy Simulations (SBES)(Menter, 2018). SAS and SBES aim for a better resolution of the turbulent structures by either decreasing the added turbulence modeling or relying on LES turbulence models. They both rely on shielding functions to delimit volumes where a "close-to" LES formalism is used.

For k-$\omega$ and SAS, the numerical schemes used are second-order in space and second-order backward Euler in time. For SBES, the spatial numerical scheme is changed to a blended Central Difference Scheme (Leonard, 1991; Jasak et al., 1999),
switching between second-order and first-order up-wind based on the local vel

### 1.3.1    Ansys CFX Grids

For the CFX grid, a combination of structured and unstructured grids is used, to keep more control near the blade while exploiting the unstructured expansion of the grid further away. The different meshes are generated using Pointwise v22.2, allowing the control of the structured mesh. The first cell size normal to the surface is set to 1e-5m with a growth rate of
1.07 which ensures a y+ of one or less. Several grid resolutions are investigated in this work where only the structured part is refined, i.e. the background unstructured cell size is kept constant. The baseline structured mesh used, if otherwise not stated, has 500 cells spanwise, 544 cells chordwise and 152 cells normal to the surface. This adds up to $\approx$50M cells total - 48M in the structured region and 2M in the unstructured region. The mesh quality is evaluated based on the cell length ratios in the chordwise and spanwise direction, keeping them below 1.5. In the structured region, near the blade, the Vertex-Centered-
Orthogonality (VCO, area-weighted average of the orthogonality angles associated with each bounding face of the dual mesh control volume around the vertex. A 90 degrees VCO represents perfect orthogonality) is kept higher than 20 degrees. For the unstructured region, a Delaunay triangulation algorithm is used and a smooth transition from structured to unstructured is achieved by using a growth rate of 1.07 for the tetrahedron. An overview of the mesh is given in Fig. 4.a. The domain is square with 500m side lengths ($\approx$ 5 blade lengths) and the blade is placed in the center. Side boundaries are velocity inlets and
pressure outlets.

To take into account the motion of the selected mode shape the mesh is deformed periodically at the mode frequency. The mesh deformation is computed only during the initialization step as the displacement is imposed. This deformation is computed



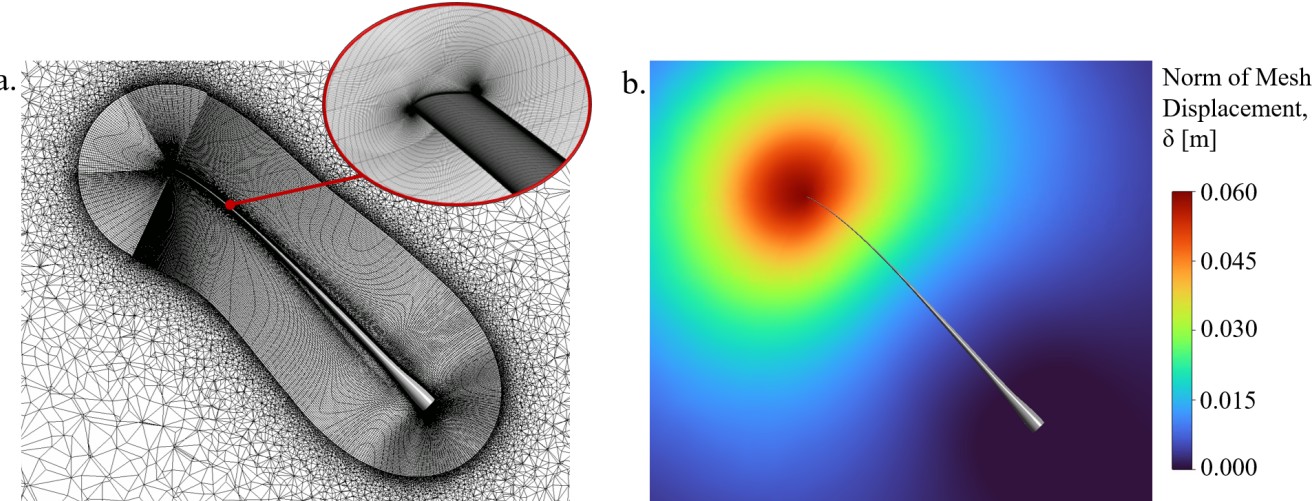

**Figure 4.** a. Near blade volume grid in CFX setup. Structured mesh is grown from the blade surface and switched to unstructured at ≈20m from the surface. b. Diffusion of the mesh displacement norm emanating from the blade surface at a given timestep.

by diffusing the displacement registered on the blade boundary to the neighbouring mesh cells. To prevent any cell from folding over, a mesh stiffness is defined. This stiffness is set to increase near the blade boundaries at a cubic rate and after a distance to the blade boundary of 1m. The obtained mesh displacement at a given timestep is depicted in Fig. 4.b. The VCO and negative cell volumes are monitored to ensure that the grid remains suited for resolving the flow.

## 1.4 Setup differences

The main difference between the two used CFD codes is their discretization methods, EllipSys3D being a structured solver, while CFX uses unstructured grids. Both of these have pros and cons; the unstructured grid approach being more flexible in terms of grid manufacturing, however, often resulting in a slower performance. In this work, the grid close to the surface was chosen to being structured for the CFX solver as well, to avoid a too rapid dissipation of the shed vortices, found when using an unstructured approach. Further from the surface, the unstructured grid, rapidly expands, limiting the cell count, i.e. ensuring faster computations. For the structured grid in EllipSys3D an expansion of cells also happens when moving far from the surface. The number of cells used also varies between the two setups, based on grid sensitivity studies, see more in Sec. 2.1. This is partly a consequence of the convective schemes having different orders of accuracy, being 4th-order accurate in EllipSys3D while being 2nd-order accurate in CFX.

Domain shapes differ between the two being spherical for EllipSys3D and square for the CFX setup. This should have no impact, as boundary conditions are far from the considered blade.

The turbulence models implemented in the two solvers differ, but should have similar capabilities of capturing the vortex shedding with varying degrees of accuracy from URANS k-$\omega$ SST (Menter, 1993) in both solvers, to the higher fidelity hybrid



RANS/LES models like IDDES (Gritskevich et al., 2012; Menter et al., 2003; Shur et al., 2008) in EllipSys3D and SAS (Egorov et al., 2010) and SBES (Menter, 2018) in CFX.

Finally, the blade surface shows discrepancies at the tip. As the meshing methodology differs between the two setups, the tip cap surface used in CFX is flat, while it is rounded in EllipSys3D. This introduces an $8\,cm$ difference corresponding to less than $0.1\%$ of the total blade length, which showed a low impact on the aerodynamic spanwise power distribution introduced in the following section.

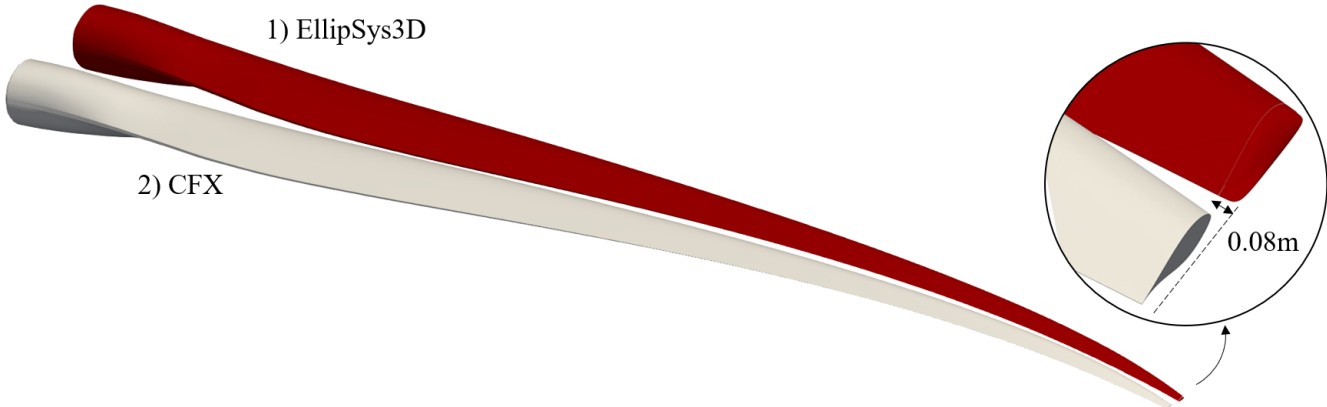

**Figure 5.** Discrepancies of the blade surface at the tip between the two setups: EllipSys3D (rounded tip) and CFX (flat tip).

## 1.5 Analysis

### 1.5.1 Aerodynamic power

In this paper, aerodynamic power will be defined as positive when injecting power into the structural system, while negative when damping the structural response. When calculating aerodynamic power, the mean power over $n$ full motion cycles is considered, see Eqs. 1 and 2. Through one motion period, there might be both positive and negative contribution of power, and so the total power over the full cycle needs to be considered to assess whether the structural response is excited or damped. The power, $P_{A,TOT}$, is total power for the full blade, meaning that power is found spanwise,$P_{A,SPAN}$, and integrated over the length of the blade.

$$P_{A,SPAN}(y) = \frac{1}{T \cdot n} \int_{t=t_0}^{t_0+T \cdot n} F(y)\dot{u}(y)dt \,, \tag{1}$$

$$P_{A,TOT} = \int P_{A,SPAN}(y)dy \,, \tag{2}$$





$t_0$ is the start time of the integration, $T$ is the time for one motion period and $n$ the number of full cycles considered. $F(y)$ and $\dot{u}(y)$ are the force and structural velocity along the blade span $y$.

When considering the risk of VIVs, it is important to realize that the total power $P_{A,TOT}$ is the driving factor. If this is
210 negative, the vibration is damped, despite there being positive values of $P_{A,SPAN}$ at some spanwise positions.

### 1.5.2  Assessment of VIV amplitudes

As with the aerodynamic power injected to the structure for a given amplitude, the corresponding power dissipated by structural damping can be found and compared to the aerodynamic power to asses whether the operating point is stable or not.

Structural damping is estimated using modal analysis. For single degree of freedom systems, the energy $E_D$ dissipated by
215 damping during one cycle of harmonic vibration of frequency $\omega$ is given by (Paz, 2012)

$$E_D = \int\limits_0^{2\pi/\omega} (c\dot{u})du = \int\limits_0^{2\pi/\omega} c\dot{u}^2 dt \tag{3}$$

Given $\dot{u}(t) = U_0\omega cos(\omega t - \phi)$ the equation (3) becomes

$$E_D = cU_0^2\omega^2 \int\limits_0^{2\pi/\omega} cos^2(\omega t - \phi)dt = \pi c\omega U_0^2 \approx 2\pi\zeta k U_0^2 \approx 2\pi\zeta\omega^2 m U_0^2 \tag{4}$$

where $\zeta = c/c_{cr} << 1$ is the damping ratio, $\omega = \sqrt{k/m}\sqrt{1-\zeta^2} \approx \sqrt{k/m}$ is the damped natural frequency and $c_{cr} = 2\sqrt{km}$
is the critical damping. Parameters $m$, $k$ and $c$ are respectively; mass, stiffness and structural damping.

To obtain the power dissipation the energy needs to be divided by the period $T = 1/f = 2\pi/\omega$

$$P = c/2\omega^2 U_0^2 \approx \omega\zeta k U_0^2 \approx \omega^3\zeta m U_0^2 = F_{STRUC}U_0^2 \tag{5}$$

An equivalent single degree of freedom system for a given mode can be constructed using modal analysis, thus, the stiffness and damping are replaced by modal stiffness and modal damping for a given mode to compute the energy dissipated when the
225 structure moves by a unit amplitude with a certain mode shape.

With eigen matrix [V] containing in each column the eigen vector $\{V_i\}$ for a given mode $i$

$$\begin{aligned}
[M_{modal}] &= [V]^T[M][V] \\
[C_{modal}] &= [V]^T[C][V] \\
[K_{modal}] &= [V]^T[K][V],
\end{aligned} \tag{6}$$





where the first edgewise eigenvector has been scaled to correspond to 1 m blade tip deflection in the edgewise direction. Thus the effective mass, damping and stiffness values to be used in equation 5 are

$$
\begin{aligned}
m_{eff} &= [M_{modal}](i,i) \\
c_{eff} &= [C_{modal}](i,i) \\
k_{eff} &= [K_{modal}](i,i)
\end{aligned}
\tag{7}
$$

It is also evident from equation (5) that structural damping is a quadratic function of the amplitude of the displacement.

Using the method above and HAWCSTAB2 (Hansen, 2004), the structural damping dissipation factor $F_{STRUC}$ of the blade undergoing the first edgewise mode was found to **495.8 W/m$^2$**. This deviates from the value of 540 W/m$^2$ that was found in previous FSI studies (Grinderslev et al., 2022). The reason for this deviation is likely a combination of numerical damping in the loosely coupled FSI framework and deviations of the aeroelastic deflection shapes observed in the FSI simulations from the purely structural modeshapes investigated here. However, considering the uncertainties in the structural damping of wind turbine blades, the agreement within 10 % is acceptable.

## 2 Results

### 2.1 Grid and time dependency

#### 2.1.1 Grid sensitivity

Various grid configurations have been tested in the present study using varying turbulence models. In the higher fidelity turbulence models (SAS,SBES and IDDES), the resolved length scale in the LES region depends on the grid cell size itself, meaning that large changes of the grid can lead to large changes of the resulting flow. Two separate grid studies were conducted. First, the EllipSys3D solver using the IDDES turbulence models with different inclination angles and largely varying grid resolutions. The grid setups tested are defined in Table 1 and go from the coarsest case, *E-A*, of $\approx$ 12M cells to the finest case, *E-E*, of 2281M cells. Obviously, a grid with more than two billion cells, is more an academic case than a practical case, due to the corresponding immense computational cost for both running and post-processing. Luckily, it is also found unnecessarily fine in the current study, which is depicted in Fig. 6, showing the total power for various flow inclinations as result of grid. For all the cases, a 1m amplitude has been imposed. As seen, the resulting power becomes close to stable from grid case *C* and finer, and the sensitivity seems to be highest for the lower inclination cases with 30° and 40° inclination. For the higher inclination cases $I = 50°$ and $I = 60°$, the sensitivity is in general low, which was also found in previous studies by Horcas et al. (2022b).

Note that for the sake of visibility, the figure has number of cells spanwise on the x-axis, despite cases *D* and *E* having varying numbers of cells in normal and chord-wise directions as well, for the sake of grid quality.

Physically, it makes sense that lower inclination cases are more sensitive to grid and turbulence model, than higher inclination cases, as the amount of chaotic natural shedding in low5 inclination cases is quite high. For higher inclinations, there seems to be much more shedding that is correlated with the motion of the blade, meaning larger more ordered vortices are resolved. For



the flow case of $I = 30°$, the importance on considering the grid is evident, as the CFD predicts positive power injection for the coarse grid setups $A$ and $B$, but negative power for the finer cases. In the specific case, this is conservative, as one would "over design" the turbine if considering it in the design. However, it is not a given that the opposite cases could not exist, where
the positive injection of power would only be captured for finer grid setups as is almost the case for $I = 40°$.

This finding indicates that the previously found VIV risk mapping from (Horcas et al., 2022b), shows false positives in the low inclination region around 30 degrees, as the grid in that study was coarser than what is here found to be necessary. However, the main risk region found in the mapping at higher angles of inclination is valid with these findings.

**Table 1.** Grid refinement cases for the EllipSys3D setup.

|  | Spanwise | Chordwise | Normal | Total number of cells |
|---|---|---|---|---|
| E-A | 160 | 256 | 256 | $\approx 12M$ |
| E-B | 256 | 256 | 256 | $\approx 18M$ |
| E-C | 512 | 256 | 256 | $\approx 35M$ |
| E-D | 1024 | 512 | 512 | $\approx 285M$ |
| E-E | 2048 | 1024 | 1024 | $\approx 2281M$ |

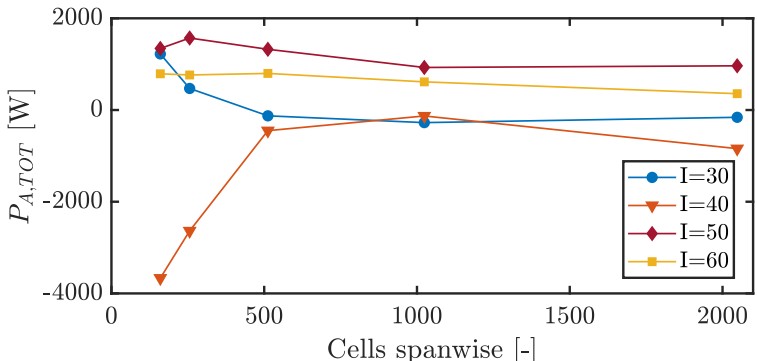

**Figure 6.** Total aerodynamic power per cycle, $P_{A,TOT}$, for varying grid refinements and flow inclination angles. $P = 100°$ for all cases.

.

Secondly, a grid study using various turbulence models was conducted using the CFX setup and the turbulence models
URANS, SAS and SBES for the inclination angle of $I = 30°$, which as mentioned was found quite grid sensitive for the EllipSys3D setup. The grid setups for the CFX cases are described in Table 2. The results are shown in Fig. 7 along with the corresponding IDDES cases from EllipSys3D.



**Table 2.** Grid refinement cases for the CFX setup. Total number of cells is given for the structured part of the mesh.

|       | Spanwise | Chordwise | Normal | Total number of cells |
|-------|----------|-----------|--------|-----------------------|
| C-A   | 250      | 272       | 152    | ≈ 11M                 |
| C-B   | 500      | 272       | 152    | ≈ 23M                 |
| C-C   | 500      | 544       | 152    | ≈ 48M                 |

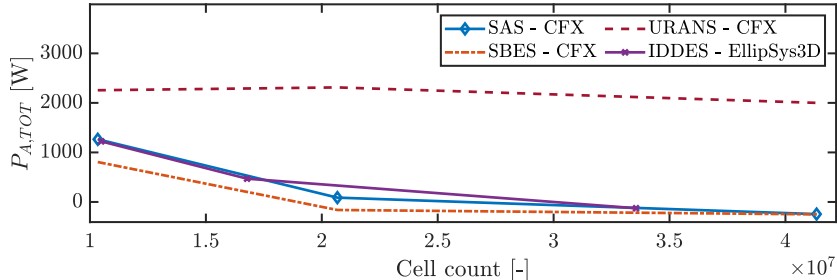

**Figure 7.** Total aerodynamic power per cycle, $P_{A,TOT}$, for varying grid refinements and turbulence models. Flow case *P100I30*

.

As seen, the behaviour of the higher fidelity models, SAS, SBES and IDDES is very similar and find a large dependency of the grid setups. The URANS case, however, does not see this dependency but appears to overshoot on the aerodynamic power injection for all grids considered.

### 2.1.2 Time step sensitivity

The sensitivity to time step size for the EllipSys3D simulations where studied for the flow case *P100I50* with a 1m amplitude. The baseline time step was set to 6e-3 seconds, which was found suitable in the earlier work by Horcas et al. (2022b) and results in 250 time steps per motion cycle for the first edgewise mode. The time step was varied to half and double the baseline, and the resulting power injection distribution along the span is given in Figure 8. It is worth mentioning that each time step has between 5 to 20 inner iterations, which dynamically changes based on the convergence of the flow residuals. It has in general been found that a deep convergence is needed to capture the power injection well. This means that various time steps could suffice if the number of inner iterations ensure a sufficiently deep convergence.

For the CFX setup, a similar sensitivity is performed. The baseline time step is less strict, reaching 1.5e-2 seconds, resulting in 100 time steps per motion cycle for the same first edgewise mode. It is then reduced by two and four, where the resulting power injection along the span is shown in Figure 8. In a similar way to EllipSys3D, the time step inner iterations is fluctuating





according to the flow residuals convergence. A limit of 5 inner iterations was used for CFX. As seen in Fig. 8, the two solvers obtain only marginally different results when using time steps half the size of the baseline choices.



**Figure 8.** Spanwise distribution of aerodynamic power injection for various time steps in the CFD solvers for flow case *P100I50*. For EllipSys3D the baseline mesh and IDDES were used. For CFX the baseline mesh and SBES were used.

## 2.2 Turbulence model dependency

### 2.2.1 High inclination

The high inclination case has flow coming with inclination angle $I$ of 50 degree and pitch angle $P$ of 100 degrees. In this scenario, the shedding is quite correlated for all turbulence models as depicted in Fig. 10, but with the highest degree of correlation





for URANS turbulence modelling, as this cannot resolve the small structure vortices. The spanwise power distribution is quite similar, no matter the turbulence model, and the accumulated power injected to the blade is also close between all methods yielding a minimum of 1280W from EllipSys3D URANS and a maximum of 1450W from CFX SAS.

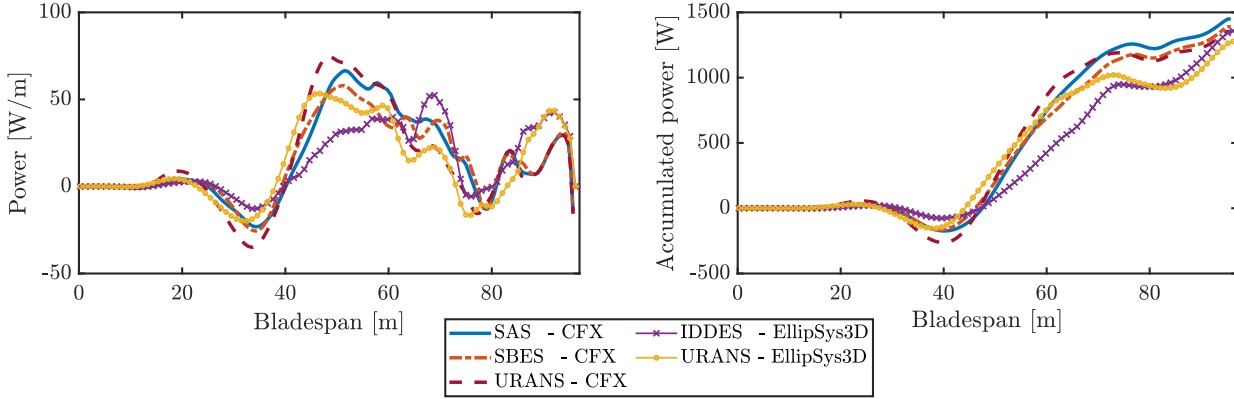

**Figure 9.** Spanwise distribution of aerodynamic power for flow case *P100I50*. Left: average power per cycle $P_{A,SPAN}$; right: $P_{A,SPAN}$ accumulated over span.




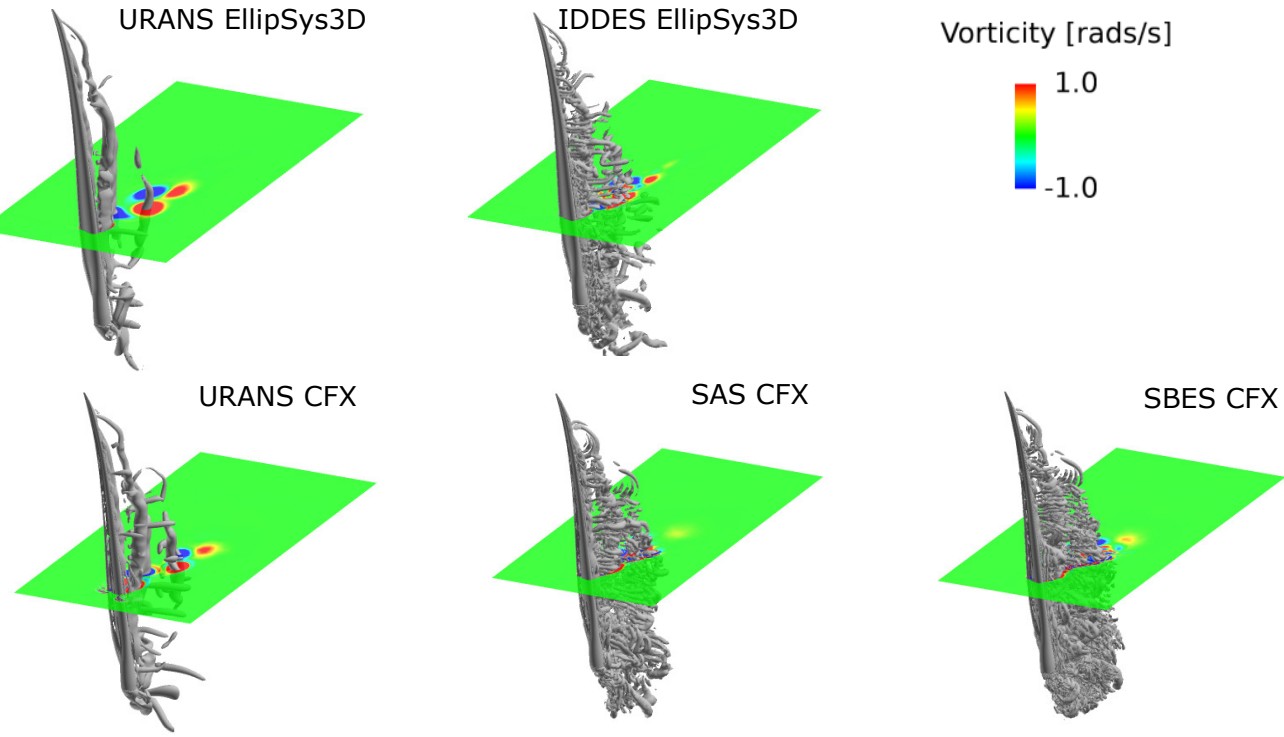

**Figure 10.** Vorticity fields resulting from the used turbulence models along with isosurfaces of q-criterion = 5.0. Flow case *P100I50*.

### 2.2.2 Low inclination

The low inclination case has flow coming with inclination angle $I$ of 30 degrees and pitch angle $P$ of 100 degrees. Here, a large degree of natural, more chaotic, shedding occurs, which is seen to be quite different between the various turbulence models, see Figs. 11 and 12. For IDDES, SAS and SBES a high degree of small scale vortices are shed from the blade, without large

scale spatial or temporal correlation.





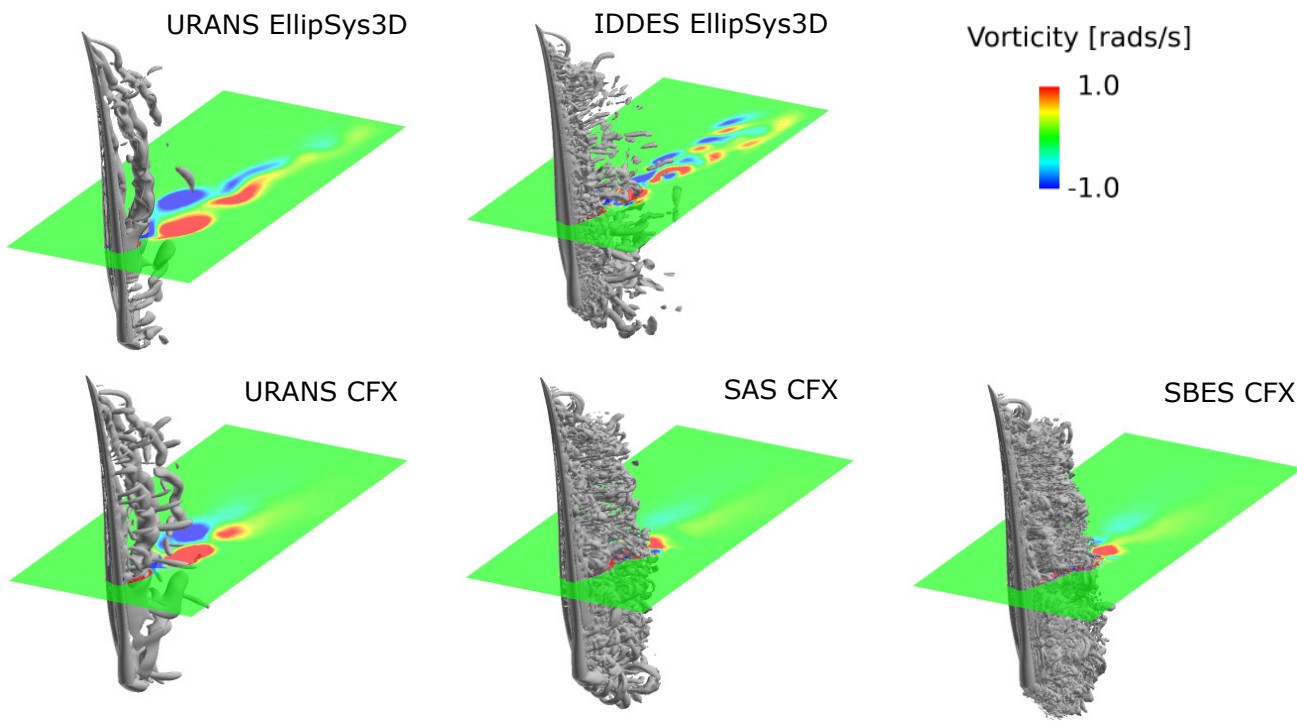

**Figure 11.** Vorticity fields resulting from the used turbulence models along with isosurfaces of q-criterion = 5.0. Flow case *P100I30*.

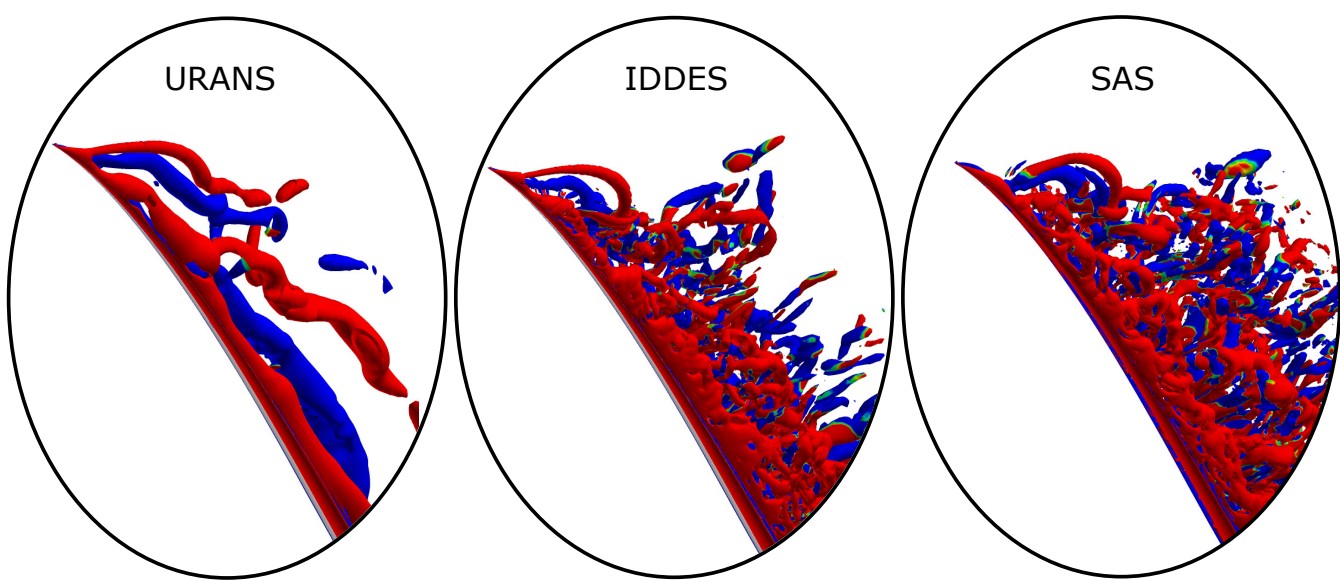

**Figure 12.** Zoom in on outer part of the blade for URANS (DTU setup), IDDES and SAS turbulence. Q-criterion = 20.0 and isosurfaces colored by vorticity between -1 and 1 rads/s. Flow case *P100I30*.





This shedding of small scale vortices results in negative accumulated power along the span. The spanwise power distribution is therefore also much less in agreement between the lower and higher fidelity turbulence models than in the high inclination case. The lower fidelity URANS turbulence models predict high spanwise correlation resulting in a high power injection between 1560W and 2000W. The higher fidelity models, however, predict the situation to be positively damped with

an accumulated power of ≈-300W.

It is important to note that for IDDES, SBES and SAS, this flow case resulted in positive power injection for lower grid resolution - presented in Figs. 6 and 7 depicting the evolution of total aerodynamic power injection according to the grid resolution and turbulence models for this low inclination. In Fig. 7, the URANS seems converged numerically, yet presents results in direct opposition to the one of SBES, SAS and IDDES once grid convergence is achieved. This is likely due to the

fact that these higher fidelity models are blending URANS and LES models, and with lower grid resolution, the URANS region is larger meaning the spanwise vortex shedding becomes more correlated for coarser grid resolutions. This shows that URANS, irrespectively of the grid resolution, is not suited for simulating VIV of blades with low inclination flow.

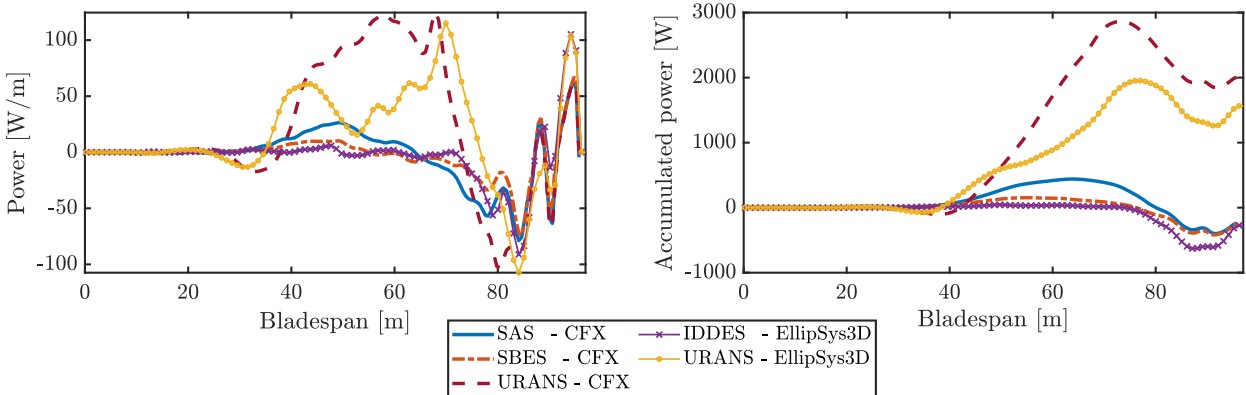

**Figure 13.** Spanwise distribution of aerodynamic power for flow case *P100I30*. Left: average power per cycle $P_{A,SPAN}$; right: $P_{A,SPAN}$ accumulated over span.

Figures 11 and 12 show the vortices shed from the blade captured by the various turbulence models. In Fig. 12 the outer approximately 40% of the blade is shown and it is clear how the URANS turbulence model creates much more coherency between the vortices than what is seen for IDDES and SAS, which both create more incoherent natural shedding.

### 2.3    Resulting vibrations

Using both CFD setups, sweeps of amplitudes up to 2m were conducted using the various turbulence models. URANS and IDDES simulations were conducted using the EllipSys3D setup, while SBES and SAS simulations were performed on the CFX

setup. By these sweeps an approximation of the vibration level can be given, using the structural damping of the considered blade. As presented in Sec. 1.5.2, the power dissipated by structural damping is proportional to the square of the vibration



amplitude. For the considered blade a relation between dissipated power and amplitude was found to: $P_{STRUC} = 495.8 \cdot A_{tip}^2$. By this relation, an effective power $P_{EFF} = P_{A,TOT} - P_{STRUC}$ can determine whether the blade is in a stable or unstable situation, as depicted in respectively green and red regions of Fig. 14.

As seen, for the high inclination case with $I = 50°$ a similar trend of effective power is found between the various models varying mostly at higher amplitudes. The equilibrium points of said simulations lie between $\approx$1.25m for SAS and $\approx$1.6m for IDDES. For the low inclination case, however, it is again clear that the URANS model results in far from realistic scenarios due to the artificial vortex coherency created. For this flow scenario, all higher fidelity simulations (IDDES, SBES and SAS) lie well within the stable region, meaning that no vibrations should occur. URANS simulations, however, show high injection of power all the way to 1.25m amplitude. One could state that this is acceptable as the result is conservative, but note that this was

not the case for the high inclination case. A scenario in which higher fidelity turbulence will predict vibrations where URANS turbulence does not, is plausible. However, this scenario, if present, is still to be revealed to the authors.

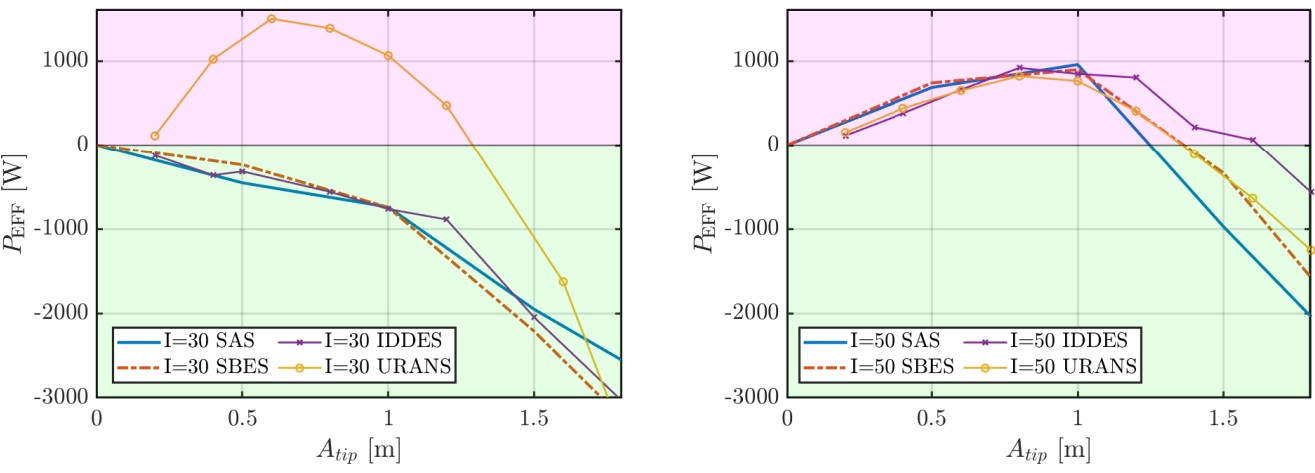

**Figure 14.** Effective power, $P_{EFF} = P_{A,TOT} - P_{STRUC}$, for various amplitudes using SAS, SBES, IDDES and URANS turbulence modeling. Flow cases *P100I30* (left) and *P100I50* (right)

## 3   Conclusions

A comprehensive study has been conducted, investigating the impacts of various simulation choices for vortex-induced vibrations of wind turbine blades. Common for all studies was the forced motion CFD approach, where the structural first edgewise

mode was imposed as a motion in the CFD simulations, which in earlier work has been shown feasible. Two independent CFD methodologies were used being the DTU in-house EllipSys3D solver and the commercial Ansys CFX solver used at Siemens Gamesa. By this, various grid strategies and turbulence models are tested and compared showing a high degree of sensitivity for especially low inclination flow, meaning spanwise flow closer to perpendicular to the span rather than along the span. It is





found that for these inclinations care is needed for the selection of turbulence model and grid. The observed differences are
due to the artificial coherency in the vortex structures created by unsteady RANS models, leading to high input of aerodynamic
power. For coarse grids, the URANS region of the higher fidelity DES-like turbulence models become too big, leading to similar results as for pure URANS simulations. For finer grids, the higher fidelity models resolve the more chaotic smaller scale
vortices, which breaks the coherence and power injection. For higher inclination cases, the sensitivities to grid and turbulence
models is much lower, as the degree of chaotic natural shedding is low compared to the coherent structures, which can be
resolved fairly well even using URANS turbulence.

This leads to the main conclusion of the present study: A lot of care needs to be taken when simulating vortex-induced
vibrations of wind turbine blades. Various conditions will need separate sensitivity investigations in order to ensure the accuracy
of the results. This is important since it is otherwise risked that much too heavy computations are conducted for cases that do
not need it. Even worse, the computations that were found to be well resolved for one case may fail to predict the VIVs in other
cases.

## 4 Future studies

The topic of VIVs becomes increasingly relevant with the increasing sizes of wind turbines, and much more research is needed.
As a continuation of the current study, an expansion on the parametric space is needed to make final conclusions on turbulence
models and grid requirements. This study shows that the necessity of high fidelity turbulence and fine grids is highly dependent
on flow scenario. As of now, no general rule of thumb about how and when to use various models can be justified. This would
need a larger mapping of flow cases and rotor designs. However, the tendency seems to be that the need of higher accuracy
increases with the degree of natural shedding along the blade, meaning that low inclinations are more difficult to compute
correctly.

Even though forced motion simulations have the possibility of being more efficient than fully coupled FSI simulations, the
simulation time needed for broad mappings is still high; especially if various amplitudes or flow velocities are needed. By use
of reduced order modeling, the needed amount of simulations could possibly be reduced significantly.

The current study only considers clamped single blades undergoing the 1st edgewise blade mode vibration. As wind turbines
are coupled systems, the coupled rotor modes should likewise be studied. This highly increases the complexity as the modes
of a wind turbine are many, and the motion of the individual blades will then depend on azimuth position.

## Acknowledgments

This work has been supported by the PRESTIGE project (J.no. 9090-00025B), granted by Innovation Fund Denmark. Computational resources were provided by the DTU Risø cluster Sophia (DTU Computing Center, 2021).





**Data availability statement**

The geometry and the structural model of the considered wind turbine are publicly available. However, fluid solvers are li-
365 censed. The data that support the findings of this study are available upon reasonable request to the corresponding author.

**Author contributions**

C. Grinderslev conducted EllipSys3D simulations, did analysis of the results, and did the main writing of the paper. F. Houtin-
Mongrolle did the majority of the CFX calculations, analysis, and methodology implementation in CFX as well as participating
in the redaction and reviewing of the paper. N. N. Sørensen conducted EllipSys3D simulations, took part in the analysis
and reviewing the paper. G. R. Pirrung conducted HAWCSTAB2 analysis, damping estimations, supported on analysis of
EllipSys3D simulations and reviewed the paper. Valuable support was provided by P. Jacobs, A. Ahmed and B. Duboc regarding
the analysis of CFX results, methodology implementation in CFX, and reviewing the paper. All authors contributed in editing
the paper.

**Competing interests**

The authors declare that there are no competing interests.





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
