# Peer review of "Forced motion simulations of vortex-induced vibrations of wind turbine blades - A study of sensitivities"

_Wind Energy Science, 2023_

## Author Comment (AC1)

**Answer to reviewer 1**

Christian Grinderslev[1], Felix Houtin-Mongrolle[2], Niels Nørmark Sørensen[3], Georg Raimund Pirrung[3], Pim Jacobs[2], Aqeel Ahmed[4], and Bastien Duboc[4]

[1]Department of Wind and Energy Systems, Technical University of Denmark, 2800, Kgs. Lyngby, Denmark
[2]Siemens Gamesa Renewable Energy, Prinses Beatrixlaan 800, 2595BN Den Haag, Netherlands
[3]Department of Wind and Energy Systems, Technical University of Denmark, Risø Campus, 4000, Roskilde, Denmark
[4]Siemens Gamesa Renewable Energy, 685 Avenue de l'Université, Saint-Etienne-du-Rouvray, 76801, France

**Correspondence:** Christian Grinderslev (cgrinde@dtu.dk)

The authors would like to thank the reviewer for the thorough review and many good points made, which will improve the paper significantly. In the following, the reviewers comments will be given in black font, while the answers from authors are given in blue.

The manuscript presents a sensitivity analysis of different CFD model fidelities and parameters to simulate VIV of a wind turbine blade under forced motion. The results obtained with two CFD models (EllipSys3D, Ansys CFX) and 5 turbulence models are compared for both low and high inclination angles. VIV is a topic of increasing interest for wind turbine blades of increasing lengths and different modelling approaches exist in the literature. As such, the present work is of relevance to the field. The methodology is also scientifically sound. The main outcome of the paper is that URANS is not suitable for simulating VIV of blades with low inclination angles.

The authors agree that one main outcome of the paper is that URANS is not suitable for VIV simulations at low inclination angles, but would also like to additionally emphasize that another finding is the significant sensitivity to grid sizes for the scale resolving turbulence models. This is something that does not seem to be well known in the field.

Although the work is interesting and relevant, the main weakness of the manuscript is that the scope of the work, and discussion of the results, is rather limited for a paper on its own. Also, the choices made in the work are not always justified.

I suggest addressing the following comments/suggestions in the revisions of the manuscript.

- The present approach of prescribing the VIV motion is justified based on previous work using a similar setup. However, in this work, it is also shown that some of the previous work was using insufficient spatial resolution to properly resolve the flow phenomena and associated effective power. Can these work still support the idea that forced motion is realistic enough? The previously published work comparing the forced motion with coupled FSI simulations (Grinderslev et. al 2022) was done with high inclination angles only, and by that the spatial resolution was sufficient, as shown in the present paper. Through the PRESTIGE project, many simulations with both FSI coupling and forced motion for various spatial resolutions and both high and low inclination angles have been conducted on various wind turbine designs, and it's been found in general that the forced motion method for all tested VIV cases works very well. This is due to the fact that all work has been done with fairly low wind speeds (5-30 m/s), meaning that the effect of the aerodynamic forces on the mode shapes and natural frequencies is low. For other phenomena, such as stall-induced vibrations at higher wind speeds or in the case where the fluid density is much higher,

the forced motion approach is likely not as good an option as the resulting mode shapes and frequencies will be significantly altered.

- Line 141: no checks for orthogonality, why is that? The reason for not doing checks on the orthogonality during runtime, is that in EllipSys3D the orthogonality does not affect the accuracy of the solution, as long as the solution is still able to converge deeply. The effect of very skewed cells could, however, be seen in numerical instabilities, which could make simulations crash. The current orthogonality correction of the normals close to the surface with a blending into the original mesh farther from the surface, works very well, even for shape optimization problems.

- The mesh sizes used is both codes are different. This is briefly described in Sec 1.4. However, it's still not fully clear why the first cell size is larger with CFX whilst it is mentioned that its numerical schemes are of lower accuracy. Please explain in more details. In Sec 2.1.2, it is mentioned that a limit of 5 inner iterations is used for CFX. This is also smaller than with the other code. This is not justified nor explained. These choices are made to fit usual choices for the specific solvers. EllipSys3D, used at DTU, works very efficiently, and a very small first cell height does not lead to convergence issues and do not add significantly to the computational time. This is not the same for Ansys CFX, which struggles more with the very thin cells near the surface. That being said, the very low first cell height used for the EllipSys3D setup is an overkill, as the y+ is much lower than 1, but it is the common choice for blade resolved simulations in EllipSys3D. The amount of used subiterations is based on experience by the specific users for the two solvers on getting the residuals low efficiently. In EllipSys3D it is often chosen to have a high maximum number of subiterations, however, if set up correctly, these will only be used for the first short part of the simulation to get residuals low and then afterwards, for the remainder of the simulation, less subiterations are needed per time step. The reason for the Ansys CFX setup working well with a higher time step as well as lower allowable subiterations, is likely due to the mesh differences and the computational schemes. However, since EllipSys3D is computationally very efficient, it has not been found necessary to investigate this further.

- Line 234 mentions the numerical damping in loosely coupled FSI. Can the authors further quantify this, based on this work or previous work? When first developing the FSI framework at DTU, Heinz (2013) studied various coupling strategies and their impact on energy conservation and loads. It was concluded that the loose coupling scheme was sufficiently good for wind energy purposes, due to the large ratio between structural mass and mass of the surrounding air, and the small time steps used in these simulations. The authors believe that the difference between HAWCSTAB2 results and former FSI results is due to postprocessing, as the aforementioned FSI results were found by analysis of multiple time responses and fitted for a good overall match see (Grinderslev, 2022). HAWCSTAB2 results are also based on purely structural mode shapes, while the FSI results are aeroelastic, which where found to be similar to the structural ones for this wind speed but not exactly the same.

- The figure legends (e.g. Fig 7) mention case names such as P100I30. This only becomes clear later in the manuscript (i.e. at the start of Section 2.2.1). It would be beneficial to introduce this naming in the text or in a table at an earlier stage in the manuscript. This is a very good point, the naming is now introduced earlier in the text, where the difinition of inclination and pitch are given as well.

- In Sec 2.1.2, the setups are presented but the results shown in Fig 8 are not really discussed. The EllipSys3D results are also not monotonically converging with decreasing time step. This deserves further explanation. A figure has been added with

the accumulated power in time, in order to emphasize better the time dependency, which also looks more as expected between the lower time steps. The non-monolithic behaviour in the average accumulated power, is likely due to the averaging over a cycle and the very non-linear behaviour of the physics. Power is a very hard parameter to converge, due to it being a product of both force and the phase between motion and force. A small change in the latter have a high effect on the total power. For this reason a higher relative uncertainty, than for instance a usual force analysis, has been accepted here. In terms of vibration assessments, the uncertainty in injected aerodynamic power is low compared to the uncertainty in actual structural damping of the wind turbine blades. More comments on this have been added to the manuscript.

Minor changes.

- Line 76: add reference at end of sentence. References added

- Line 90 of 18m/s –> up to 18m/s? Changed to "for a wind speed of 18 m/s". We have studied other winds speeds as well with the same conclusion, but those results are not published.

- Line 155: unfinished sentence corrected

- Line 255: low5 (typo) corrected

Once again, thank you very much for you comments and suggestions to the article!

**Answer to reviewer 2**

Christian Grinderslev[1], Felix Houtin-Mongrolle[2], Niels Nørmark Sørensen[3], Georg Raimund Pirrung[3], Pim Jacobs[2], Aqeel Ahmed[4], and Bastien Duboc[4]

[1]Department of Wind and Energy Systems, Technical University of Denmark, 2800, Kgs. Lyngby, Denmark
[2]Siemens Gamesa Renewable Energy, Prinses Beatrixlaan 800, 2595BN Den Haag, Netherlands
[3]Department of Wind and Energy Systems, Technical University of Denmark, Risø Campus, 4000, Roskilde, Denmark
[4]Siemens Gamesa Renewable Energy, 685 Avenue de l'Université, Saint-Etienne-du-Rouvray, 76801, France

**Correspondence:** Christian Grinderslev (cgrinde@dtu.dk)

The authors would like to thank the reviewer for the thorough review and many good points made, which will improve the paper significantly. In the following, the reviewers comments will be given in black font, while the answers from authors are given in blue.

The article describes forced motion simulations of a clamped wind turbine blade for two different inflow conditions and with two different computational fluid dynamics codes. Several turbulence models, grid resolutions and time step sizes are investigated, with analysis of the aerodynamic power on the blade. The article is thus a sensitivity study using existing codes, but provides interesting guidelines for vortex-induced vibration simulations of wind turbine blades.

Line 69-70: The forced motion (or forced response) method has also been used extensively for gas turbines, with references going back at least 20 years. It would be good to mention that this type of method only works well if the fluid does not modify the vibration frequency significantly compared to in-vacuum, so typically only for air and gasses. This is indeed correct, and even for air the assumption only works for low wind speeds, otherwise the aeroelastic mode shapes and frequencies vary too much from the structural ones. This has been emphasized in the introduction to the forced motion method Section 1.1

Line 117: The time discretization scheme for EllipSys3D seems to be missing. EllipSys uses an implicit 2nd order backward iterative time-stepping (or dual time-stepping) method. In each global time-step the equations are solved in an iterative manner, using under-relaxation. This information has been added to the paper.

Line 127-139: The grid deformation technique is described in some detail, but the text is still vague and I would not be able to reproduce the results based on this text. Can references be added? The tuning of the parameters is not documented. Some extra description including the tuning parameters of the blending has been added to the description. There is yet no specific reference to the deformation routine of EllipSys3D, but some extra explanations can be found in the thesis of Antariksh C. Dicholkar. (https://doi.org/10.11581/dtu.00000240) part II sec 1.1.2.

Line 159: Why is the first cell height 10x larger for CFX than for EllipSys3D? The flow conditions are the same and similar turbulence models are applied. The cell size used in the EllipSys3D setup is indeed unnecessarily low, as the y+ is much lower than 1 due to the low flow velocity. The reason for the choice, is that the cell height is the common practice for operational cases in EllipSys3D, where it is needed. It does not add significantly extra computational cost as EllipSys3D is very efficient on long stretched cells. This is not in the same way the case for Ansys CFX, which gain more by increasing the cell height, though

still within the limit of having y+ < 1. Admittedly, it would have been more obvious to also use 1E-5m for the EllipSys3D setup as well, however the authors deem that the consequence is low.

Line 207: How many periods (n) were used to calculate the total aerodynamic power? How does this quantity change as a function of the number of periods? Were the first periods omitted from the averaging? Only the fully converged results were used to calculate the power, and tests were made by changing the number of periods to ensure that the average per cycle did not change significantly. The number of full cycles needed to obtain a converged average power, varies especially with inclination angle, as high inclination angles often have more steady state vortices and thus require less cycles than the lower inclination cases. In all cases, at least five cycles were used for the averaging. This information has been added to the manuscript.

Line 276-278: The effect of the convergence tolerance is mentioned, but not clearly described. Show in a table the effect of the tolerance on the result. The dependence on convergence criteria has not been documented or analysed in a structured manner, and would require an excessive number of extra simulations to do. Instead the statement is based on experiences gained through a vast amount of VIV simulations during the past years during the PRESTIGE project and related activities. For EllipSys3D it has been found through experience, that reducing the normalized residuals by five orders from the initial guess of constant flow is sufficient.

Line 282: The result of a time step size study is preferably a time-varying quantity, instead of a time-average. A figure of the total power over time for the various time steps has been added to the figure.

Line 292: Figure 9 is not mentioned in the text. Reference added

Line 294: The text indicates as if the small scale vortices are shed directly from the blade, while they typically develop in the wake. Please clarify whether these small scale vortices are present already very close to the blade (so shed by the blade) or whether they develop in the wake, from larger vortices that become unstable (so develop rather than shed). This is a very good point. The vortices are created in the near wake of the blade from the shear layers that occur at the leading and trailing edges. You are right that the vortices are not created along the surface. This has been corrected in the text.

Line 317: Figure 13 is not mentioned in the text. Reference added

Minor comments:

Line 38: vortex => Vortex Corrected

Line 123: surface.The => surface. The Corrected

Caption Figure 3: clarity => clarity. Corrected

Line 128: exploring => exploiting Corrected

Line 155: Incomplete sentence Corrected

Line 208: force => spanwise force density Corrected to "spanwise force distribution" which is our usual term.

Line 222: $F_{STRUC}$ is only defined further, below Equation (7). Furthermore, this symbol F was previously used for force density, so confusing to use it also as dissipation factor. $F_{STRUC}$ is a naming chosen in a former related paper (Grinderslev et al. 2022), and for that reason it is chosen to keep the naming here to be consistent. The factor has now been introduced after Equation (5), where it is used, to avoid confusion.

Line 228: 5 => (5) Corrected

Line 242: SAS,SBES => SAS, SBES Corrected

Line 255: low5 => low Corrected

Line 316: $U_0$ was used for amplitude, now $A_t ip$, so maybe use $U_{0,tip}$. Changed $U_0$ to A instead so that the convention from (Grinderslev et al 2022) is kept.

65    Once again, thank you very much for you comments and suggestions to the article!

[revised manuscript text omitted]
\underline{U_0}\underline{A}^2 \approx 2\pi\zeta k\underline{U_0}\underline{A}^2 \approx 2\pi\zeta\omega^2 m\underline{U_0}\underline{A}^2 \tag{4}$$

where $\zeta = c/c_{cr} << 1$ is the damping ratio, $\omega = \sqrt{k/m}\sqrt{1-\zeta^2} \approx \sqrt{k/m}$ is the damped natural frequency and $c_{cr} = 2\sqrt{km}$ is the critical damping. Parameters $m$, $k$ and $c$ are respectively; mass, stiffness and structural damping.

To obtain the power dissipation the energy needs to be divided by the period $T = 1/f = 2\pi/\omega$.

$$P = c/2\omega^2 U_0 A^2 \approx \omega\zeta k U_0 A^2 \approx \omega^3 \zeta m U_0 A^2 = F_{STRUC} U_0 
[revised manuscript text omitted]